# Serum Metabolomic Profiling of Patients with Non-Infectious Uveitis

**DOI:** 10.3390/jcm9123955

**Published:** 2020-12-06

**Authors:** Hiroyuki Shimizu, Yoshihiko Usui, Masaki Asakage, Naoya Nezu, Ryo Wakita, Kinya Tsubota, Masahiro Sugimoto, Hiroshi Goto

**Affiliations:** 1Department of Ophthalmology, Tokyo Medical University, Tokyo 160-0023, Japan; sardine_harbor@yahoo.co.jp (H.S.); patty.m.best@gmail.com (M.A.); naoya.nezu@gmail.com (N.N.); ryo.0623.w@gmail.com (R.W.); tsubnkin@hotmail.co.jp (K.T.); goto1115@tokyo-med.ac.jp (H.G.); 2Health Promotion and Preemptive Medicine, Research and Development Center for Minimally Invasive Therapies, Institute of Medical Sciences, Tokyo Medical University, Tokyo 160-8402, Japan; mshrsgmt@gmail.com

**Keywords:** Behҫet’s disease, sarcoidosis, Vogt-Koyanagi-Harada disease, metabolomics, liquid chromatography-mass spectrometry, biomarker, serum

## Abstract

The activities of various metabolic pathways can influence the pathogeneses of autoimmune diseases, and intrinsic metabolites can potentially be used to diagnose diseases. However, the metabolomic analysis of patients with uveitis has not yet been conducted. Here, we profiled the serum metabolomes of patients with three major forms of uveitis (Behҫet’s disease (BD), sarcoidosis, and Vogt-Koyanagi-Harada disease (VKH)) to identify potential biomarkers. This study included 19 BD, 20 sarcoidosis, and 15 VKH patients alongside 16 healthy control subjects. The metabolite concentrations in their sera were quantified using liquid chromatography with time-of-flight mass spectrometry. The discriminative abilities of quantified metabolites were evaluated by four comparisons: control vs. three diseases, and each disease vs. the other two diseases (such as sarcoidosis vs. BD + VKH). Among 78 quantified metabolites, 24 kinds of metabolites showed significant differences in these comparisons. Four multiple logistic regression models were developed and validated. The area under the receiver operating characteristic (ROC) curve (AUC) in the model to discriminate disease groups from control was 0.72. The AUC of the other models to discriminate sarcoidosis, BD, and VKH from the other two diseases were 0.84, 0.83, and 0.73, respectively. This study provides potential diagnostic abilities of sarcoidosis, BD, and VKH using routinely available serum samples that can be collected with minimal invasiveness.

## 1. Introduction

Uveitis is a sight-threatening intraocular inflammation, characterized by a heterogeneous clinical presentation. This condition results from a complex interaction among multiple genetic, immunity, environmental, and epigenetic factors. The diagnosis of uveitis is often challenging because many diverse diseases with similar clinical features can cause this condition. These diseases are a common cause of blindness, collectively accounting for 10–25% of the current legally defined blindness worldwide [1,2]. They have a profound negative effect on the quality of life. Behҫet’s disease (BD), sarcoidosis, and Vogt-Koyanagi-Harada disease (VKH) are the three most common non-infectious diseases with uveitis [3,4,5,6]. These highly sight-threatening diseases develop in genetically susceptible individuals. The uveitis seen in these diseases is characterized by an exaggerated intraocular immune response against antigens interacting with CD4 T cells (Th1 and Th17 cells) and antigen-presenting cells. This immune response ultimately leads to alterations in blood-retinal barrier function, retinitis, choroiditis, and tissue damage [7,8,9,10,11]. The etiology of these kinds of uveitis remains a conundrum, and there is no established diagnostic biomarker. They are mostly diagnosed according to clinical symptoms and the clinical criteria shared by other types of uveitis. Diagnostic biomarkers specific for BD or VKH have not yet been identified, and lumbar puncture is a relatively invasive procedure and may not be practical or suitable for the diagnosis of VKH. Previous studies have assessed whether candidate biomarkers, including angiotensin (ACE) and soluble interleukin (IL) -2 receptor, can diagnose ocular sarcoidosis [12,13]. The results of these studies indicate that when these markers are used alone, the sensitivity and specificity of detection are insufficient to reliably identify sarcoidosis patients with uveitis. Clinical confusion can lead to the diagnosis being missed or delayed, which increases the risk of impaired vision. Thus, diagnosing BD, sarcoidosis, or VKH patients is often challenging for non-uveitis specialists. Furthermore, 80–90% of BD or VKH patients do not completely meet the clinical criteria and are described to have an “incomplete” or “probable” case of the disease, which contributes to clinical confusion and delayed diagnosis [14,15]. Treatment with systemic steroids is effective for sarcoidosis and VKH, but not BD. Since the therapeutic strategies for the three diseases are different, accurate diagnostic biomarkers are essential to improve clinical outcomes, avoid unnecessary therapies, and suppress retinal damage as well as systemic manifestations. Given the high pathological and therapeutic difference among these three diseases, identification of metabolomic biomarkers specifically expressed in the sera of corresponding patients might greatly facilitate diagnosis. Metabolomics is an -omics approach to quantify metabolites in biological samples. These small molecules frequently reflect the pathology of various diseases, especially during inflammation. Therefore, simultaneous assessment of various pathways with multiple multivariate analyses have been used to characterize the metabolic statuses of patients and diagnose the corresponding diseases accordingly [16]. The metabolites in the vitreous fluid detected using nuclear magnetic resonance (NMR) along with a partial least squared-discrimination analysis (PLS-DA) can discriminate between lens-induced uveitis and chronic uveitis [17]. However, invasive sampling of aqueous humor is required. Conversely, blood is one of the biofluids that can easily be sampled, enabling frequent testing. Previously, gas chromatography-mass spectrometry with a PLS-DA has been used to discriminate BD patients from healthy controls (HCs) [18]. NMR with principal component analysis (PCA) has been used to analyse serum samples to discriminate sarcoidosis patients from HCs [19]. Plasma samples collected from VKH patients have been analysed using liquid chromatography (LC)-MS with PCA and orthogonal partial least square discriminant analysis (OPLS-DA) to differentiate these patients from HCs [20]. All these studies have aimed to distinguish patients with a particular type of uveitis from HCs, whereas methods to differentiate different types of uveitis patients from each other have not been investigated.

Here, the serum metabolites of BD, sarcoidosis, and VKH patients were compared to identify disease-specific diagnostic biomarkers. We quantified hydrophilic metabolites using LC-time-of-flight (TOF)-MS and evaluated the discrimination abilities of these quantified data.

## 2. Materials and Methods

### 2.1. Patients and Diagnosis

Patients with active uveitis were retrospectively identified from the medical records of Tokyo Medical University Hospital between January 2016 and December 2018. Patients who had been treated with systemic corticosteroids, immunosuppressive drugs, and antimetabolites within the past 6 months leading up to the acquisition of the serum sample were excluded. Patients who had been treated with topical steroids and mydriatics were included for analysis. A total of 54 patients comprising 19 with BD, 20 with sarcoidosis, and 15 with VKH were included in the study. Sixteen subjects (7 cataract patients and 9 healthy subjects) with no history of autoimmune disease or malignant tumor were included as controls.

BD was diagnosed according to the diagnostic criteria reported by the Japanese Ministry of Health, Labor and Welfare Designated Disease Research Group [21]. Sarcoidosis was diagnosed according to the diagnostic criteria revised in 2019 [22], and VKH according to the international diagnostic criteria [23]. 

Written informed consent was obtained from all the participants of the study. The study was approved by the Ethics Committee of the Tokyo Medical University Hospital, Tokyo, Japan (project identification code: 2017-194, date of approval: 31 October 2017).

### 2.2. Serum Sample Collection

Venous blood was collected from patients with BD, sarcoidosis, or VKH and then incubated for 30 min at room temperature (24 °C). Subsequently, the serum was collected by centrifugation for 15 min at 1000× *g* and stored at −80 °C until needed for metabolomic analyses. The patients were confirmed to have active disease at the time when serum samples were collected for analysis.

### 2.3. Sample Processing

For the positive mode, a human serum sample (10 μL) was mixed with methanol (90 μL) containing 149.6 mM ammonium hydroxide (1% (*v*/*v*) ammonia solution) and 1.5 μM of each standard (d_8_-spermine, d_8_-spermidine, d_6_-*N*_1_-acetylspermidine, d_3_-*N*_1_-acetylspermine, d_6_-*N*_1_,*N*_8_-diacetylspermidine, d_6_-*N*_1_,*N*_12_-diacetylspermine, hypoxanthine-^13^C_2_,^15^N, and 1,6-diaminohexane). Following centrifugation at 20,380× *g* for 10 min at 4 °C, 90 μL of the supernatant was transferred to a fresh tube and vacuum-dried. The sample was reconstituted with 90% methanol (10 μL) and water (190 μL), then vortexed and centrifuged at 20,380 × *g* for 10 min at 4 °C. For LC–MS analysis, 1 μL of each supernatant was injected into the system.

For the negative mode, a human serum sample (10 μL) was mixed with methanol (90 μL) containing 1 μM of the standard (camphor-10-sulfonic acid). After centrifugation at 20,380× *g* for 10 min at 4 °C, 90 μL of the supernatant was transferred to another tube and vacuum-dried. The sample was reconstituted with 90% methanol (10 μL) and water (190 μL) containing 20 μM of each standard (sulphanilic acid and methionine sulphone), and then vortexed and centrifuged at 20,380× *g* for 10 min at 4 °C. The samples (1 μL per supernatant) were then injected into the LC–MS system. 

### 2.4. LC-MS Systems

The LC-MS instrument has previously been described in detail [24]. Briefly, Agilent Technologies 1290 Infinity LC system and G6230B time-of-flight MS (TOF–MS) (Agilent Technologies, Santa Clara, CA, USA) were used. Each sample was analysed in the positive and negative modes. The conditions for the analysis at the positive mode were set as described previously with slight modification [25]. The temperature of the LC columns was set at 40 °C. For the negative mode, the chromatographic separation was performed using an ACQUITY HSS T3 column (2.1 i.d. × 50 mm, 1.8 μm; Waters, Milford, MA, USA) at 30 °C. The mobile phase consisting of solvent A (0.1% formic acid in water) and solvent B (acetonitrile) was delivered at the flow rate of 0.3 mL/min. The gradient elution is detailed in Appendix A. The total run time of the LC–MS analysis was 11 min per sample. The MS setting at the negative mode was as previously described with slight modification [25]. In this study, 50–1200 *m*/*z* was used.

### 2.5. Data Processing

Annotation tables for both positive and negative modes were generated based on the results of the LC-MS analysis of 166 standard compounds, including amino acids, polyamines, organic acids, and several metabolites. Peaks were extracted using MassHunter software according to the retention time and *m*/*z* values. All peaks were visually inspected and manually curated if necessary. Metabolites below detection sensitivity in more than half of the samples in all groups were excluded. Relative peak areas of the metabolites were calculated by dividing by the area of the internal standard and were quantified according to the relative peak areas of a standard mixture, analysed before the samples. The analysis was performed using Agilent MassHunter Qualitative Analysis software (version B.08.00; Agilent Technologies, Santa Clara, CA, USA).

### 2.6. Statistical Analysis

For these comparisons, four multiple logistic regression (MLR) models were developed and evaluated to access the discrimination abilities of multiple metabolite sets. Figure 1B shows the processing flow of development and validation of the MLR models. First, the metabolites showing a significant difference in the two groups (*p* < 0.05, Mann-Whitney test) were selected. Backward stepwise feature selection with *p* = 0.05 threshold was used to select the independent minimum number of metabolites by minimizing overfitting of the model to the observed data. To evaluate the generalization abilities of these models, *k*-fold cross-validation tests were conducted (*k* = 2, 3, 4, and 5) with 200 times of various random values for each *k* value. To eliminate optimistic results, resampling tests were also conducted. Virtual datasets were generated by random sampling allowing redundant selection and used as the validation of the developed models. This process was conducted 200 times with random values for each model. The discrimination accuracy was evaluated by the area under the receiver operating characteristic (ROC) curve (AUC).

JMP Pro. 14.1.0 (SAS Institute Inc., Cary, NC, USA), GraphPad Prism (Version 8.4.2; GraphPad Software, San Diego, CA, USA), R (ver 4.0.2, www.R-project.org) and Mev TM4 (version 4.7.4, http://mev.tm4.org) were used for these analyses.

## 3. Results

### 3.1. Summary of Quantified Metabolic Profiles

Subject characteristics are summarized in Table 1. In total, 78 metabolites were identified and quantified in serum samples using LC-TOFMS (Appendix A). These metabolites were visualized by a heatmap representation, which showed 5 distinct clusters labelled (**1**) to (**5**) (Figure 2). For example, metabolite concentrations in cluster (**1**) was higher in sarcoidosis than in the other three groups. Those in cluster (**4**) were higher in BD compared to the other groups. Concentrations of all metabolites in cluster (**2**) were higher in the control, and some of them were also higher in sarcoidosis. The metabolite concentrations in cluster (**3**) also were higher in BD while several of them were also high in control. The metabolite concentrations in cluster (**5**) were higher in VKH, while several of them were high in BD.

The metabolites showing significant differences (*p* < 0.05, Mann-Whitney) in the comparisons (**a**)–(**d**) were depicted as black boxes below the heatmap (Figure 2). Comparisons (**a**) to (**d**) resulted in 4, 14, 10, and 9 metabolites showing significant differences. For example, the comparison of sarcoidosis vs. BD + VKH (labelled (**b**)) showed the largest number of significantly different metabolites, while the comparison of control vs. three diseases (sarcoidosis + BD + VKH) (labelled (**a**)) showed the smallest number of significantly different metabolites.

The pathways to which each metabolite belong are shown as color boxes at the bottom of Figure 2. For example, cluster (4) included metabolites in glycolysis, such as pyruvate and lactate (coloured red). 

### 3.2. Discrimination Abilities of Metabolites

To evaluate the discrimination ability of multiple metabolite concentrations, MLR models were developed for comparison (**a**)–(**d**) in Figure 1A. The metabolites used for the model were selected by stepwise feature selections (Table 2). Three models included two metabolites. The model of comparison (**a**) to discriminate sarcoidosis +BD + VKH from Control included adenosine and nicotinamide. The model of comparison (**b**) to discriminate sarcoidosis from BD + VKH included thiamine and asparagine. The model to discriminate BD from sarcoidosis + VKH included thiamine and uridine. The model to discriminate VKH from BD + sarcoidosis included only a metabolite, citrulline.

The discrimination abilities of the models are summarized in Figure 3. The AUCs for comparisons (**a**) to (**d**) were 0.72 (95% CI; 0.58–0.86, *p* = 0.0072), 0.84 (95% CI; 0.74–0.96, *p* < 0.0001), 0.83 (95% CI; 0.72–0.94, *p* < 0.0001), and 0.73 (95% CI; 0.58–0.88, *p* = 0.0099), respectively. The ROC curves and the distribution of the predicted values are shown in Figure 3**A**–**H**, respectively.

To evaluate the generalization ability of these models, we conducted cross-validation and resampling tests (Appendix A). For example, the model of comparison (**a**) showed median AUCs of 0.68 (*k* = 2), 0.69 (*k* = 3), 0.69 (*k* = 3), and 0.69 (*k* = 4) for 200 tests each. The AUCs were almost constant regardless of *k* value. This trend was also observed for the other comparisons (**b**–**d**). The resample tests showed median AUC values of 0.73, 0.87, 0.84, and 0.73 for comparisons (**a**–**d**), respectively.

## 4. Discussion

This study used LC-TOFMS–based metabolomics to profile the hydrophilic metabolites in the serum samples collected from patients with BD, sarcoidosis, or VKH, or from control subjects. Among these three major forms of uveitis, biomarkers such as sIL-2R and ACE were found only in sarcoidosis [12.13]. Therefore, diagnoses of BD and VKH are usually diagnosed based on clinical features. Thus, we tried to explore novel biomarkers showing discrimination ability among these three diseases. Although several previous studies analysed serum metabolites, all these studies compared patients having a particular disease with HCs [18,19,20]. To our knowledge, this study represents the first time the three diseases are compared.

Metabolites clustered, based on the quantified concentrations, resulted in disease-specific differences (Figure 2). Although BD, sarcoidosis and VKH are all uveitis, they are completely different diseases. When we combine the three together in one group, this highly heterogeneous group is likely to exhibit a diverse metabolomic profile, resulting in low discrimination when compared with the control group. When the three uveitis diseases were compared with each other, their distinct metabolomic profiles probably increase the discrimination sensitivity among the three diseases. We mapped the pathway to which each metabolite belongs to the clustered data. Since we quantified the hydrophilic metabolites, most of the metabolites belonged to amino acids, glycolysis, tricarboxylic acid (TCA) cycle, purine, pyrimidine, and polyamine pathways. Amino acids showed different patterns in each disease. For example, metabolites in arginine and proline metabolism (light skin color) were distributed in various clusters. Aspartate, glutamate, and proline were distributed in cluster (3) which showed higher concentrations in BD. Arginine metabolism has been identified as an important pathway for establishing the responsiveness of immune cells [26,27]. Arginine induces metabolic changes between glycolysis and oxidative phosphorylation, promoting the survival of activated T-cells and the generation of central memory-like cells [28]. The changes observed in the arginine biosynthetic pathway may underscore the T-cell dysfunction reported in BD. An intermediate metabolite in the TCA cycle (light blue), malate, was distributed in cluster (4), showing higher concentration in BD. Pyruvate was also distributed close to malate (Figure 2). Immune processes have important bioenergetic and biosynthetic requirements that are met by dynamic changes in energy metabolism [29,30]. In BD, it is reasonable that resting T and B cells depend on the anaerobic glycolysis for energy metabolism and occasionally metabolize pyruvate at the TCA cycle, whereas activated T and B cells meet the bioenergy demand of the process by upregulating the aerobic glycolysis [31]. We speculate that these metabolites may contribute to the high inflammatory state and disturbed energy metabolism found in these patients. 

In this study, thiamine showed the highest levels in patients with sarcoidosis but the lowest levels in BD patients, and it was the best discriminator for both types of uveitis. Sarcoidosis is a type of granulomatous uveitis and is possibly related to *Cutibacterium acnes* (*C. acnes*), which has been suggested to be an anerobic causative bacteria and pathogenetic factor of sarcoidosis [32]. We can hypothesize that *C. acnes* is trapped in granuloma under hypoxic condition [32,33]. The observed increase in thiamine levels is considered to indicate the increased anaerobic metabolism and nutrient of *C. acnes* [34]. Interestingly, a significant decrease in thiamine levels was found in BD, which is a representative of non-granulomatous uveitis.

Glutamate was distributed in cluster (3) (Figure 2), showing a higher concentration in BD. Previous studies have demonstrated an increased concentration of glutamate in the synovial fluid of patients with BD [35]. In addition, brain astrocytes induce the release of high-mobility group box 1-mediated glutamate, and its expression is significantly increased in BD [36]. Extracellular high-mobility group box 1 levels are increased in patients with BD and intestinal involvement [37]. These observations are highly consistent with our results and explain the increase in glutamate levels in BD with uveitis. However, the pathological effects of glutamate on ocular inflammation and immune function are not fully understood.

Nicotinamide was also distributed in cluster (1) and showed significantly lower concentration in VKH (Figure 2). Nicotinamide is known to have anti-inflammatory effects, inhibiting proinflammatory cytokines such as tumor necrosis factor, interleukin (IL)-1 β, IL-6, and IL-8 [38]. It has been reported that the use of nicotinamide is effective for autoimmune disorders of the skin, including pemphigus, cicatricial pemphigoid, lichen planus pemphigoides, dermatitis herpetiformis, and immunoglobulin A bullous dermatosis [39]. Therefore, our findings suggest that nicotinamide may serve as a diagnostic and therapeutic biomarker, and nicotinamide supplementation might be an effective therapeutic strategy for VKH.

Although this study has revealed several interesting findings, selection of BD, sarcoidosis, and VKH patients from a single institute and a retrospective design are the main limitations of this study, which might cause selected bias and confounding bias. Although patients who had been treated with systemic corticosteroids within the past 6 months were excluded, some patients included in the analysis had been treated with local therapies such as topical steroids and mydriatics, which could affect the metabolomic profile. However, the numbers of BD, sarcoidosis, and VKH cases in this study can be considered relevant, given the prevalence of the disease. Nevertheless, despite the limited sample size and age distribution, serum samples from uveitis and healthy individuals could be distinguished with statistical significance. Therefore, a multi-center prospective study with more patients is warranted to confirm the present findings. This biomarker panel requires external validation through similar rigorous clinical classification before further development for clinical use. Such additional validation should be considered in a more diverse demographic group than our initial cohort. Especially, metabolomic profiling of idiopathic uveitis would be important for elucidation of novel pathogenesis of this entity and for discrimination from the three major uveitis.

## 5. Conclusions

To our knowledge, this is the first published report of a comprehensive metabolomic analysis using an LC/TOF-MS–based metabolomics approach and a blood-based biomarker panel with high accuracy for detecting BD, sarcoidosis, and VKH. Consequently, this report provides new insight into the molecular cascades underlying the pathogeneses and facilitates the specific diagnoses of the three major uveitis. Our data show that these three uveitis have distinct serum metabolite profiles. Identification of new diagnostic biomarkers may improve the clinical outcomes of uveitis cases.

## Figures and Tables

**Figure 1 jcm-09-03955-f001:**
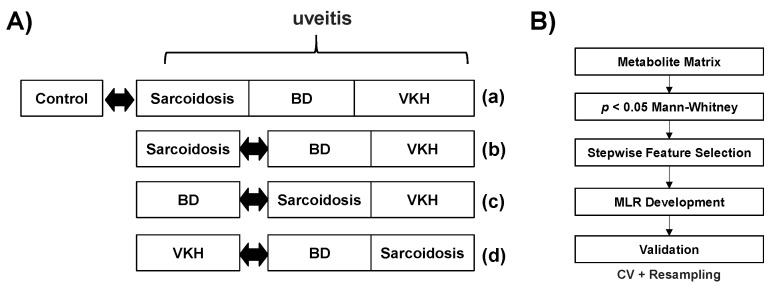
Comparative design and multiple logistic regression (MLR) model development. (**A**) Four patterns of comparison include (**a**) Control vs (sarcoidosis + BD + VKH), (**b**) sarcoidosis vs (BD + VKH), (**c**) BD vs (sarcoidosis + VKH), and (**d**) VKH vs (BD + sarcoidosis). (**B**) The development and validation of an MLR model, which starts from the metabolite concentration matrix (sample × metabolites), selecting metabolites showing the significant differences, further selecting independent minimum metabolites by stepwise feature selection, and development of an MLR model. The developed model was validated by cross-validation (CV) and resampling methods. Abbreviations: Behҫet’s disease, BD; Vogt-Koyanagi-Harada disease, VKH. The discrimination abilities of the metabolites were evaluated using both univariate and multivariate analyses. Half of the minimum concentration was substituted for the data under the lower limit of quantification. Differences in metabolite concentrations between two groups were evaluated by the Mann-Whitney U test. The design of the comparison is depicted in Figure 1A. The discrimination ability of uveitis including all three diseases from controls were evaluated (**a**). To access the diagnostic ability among three diseases, the discrimination abilities of sarcoidosis from BD + VKH (**b**), BD from sarcoidosis + VKH (**c**), and VKH from BD + sarcoidosis (**d**) were also evaluated.

**Figure 2 jcm-09-03955-f002:**
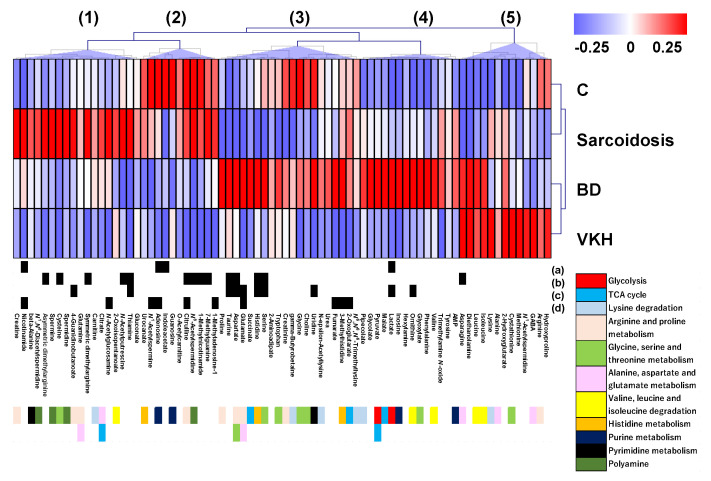
Heatmap of quantified metabolites by LC-TOFMS. Metabolite concentrations were transferred to Z-score for each metabolite. The averaged value for each group (C, sarcoidosis, BD, and VKH) were coloured in the blue-white-red scheme. Below the heatmap, black boxes are shown for the significantly different metabolites in the comparison of (**a**)–(**d**) in Figure 1. Pathways to which each metabolite belong are shown using the color box below the metabolite name. Clustering was conducted using Pearson correlation and prominent clusters are labelled (1)–(5). Abbreviations: Control, C; Behҫet’s disease, BD; Vogt-Koyanagi-Harada disease, VKH.

**Figure 3 jcm-09-03955-f003:**
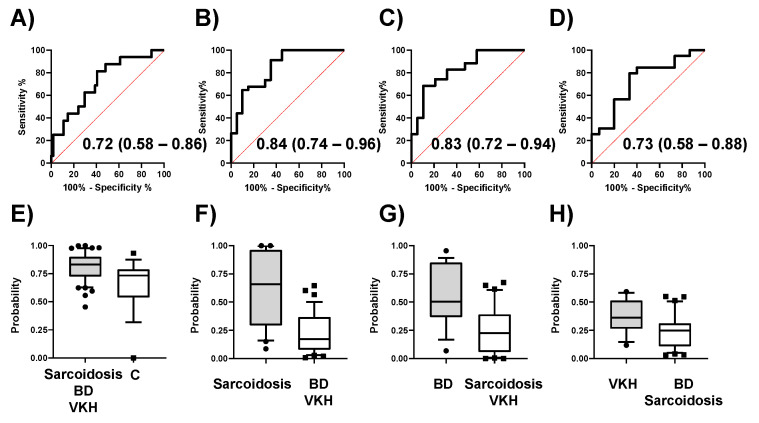
Discrimination ability of four MLR models. (**A**–**D**) Receiver operating characteristic (ROC) curves of the MLR models for comparisons (**a**–**d**) in Figure 1. (**E**–**H**) The distribution of the predicted probability of (**A**–**D**). Abbreviations: Control, C; Behҫet’s disease, BD; Vogt-Koyanagi-Harada disease, VKH.

**Table 1 jcm-09-03955-t001:** Clinical Information and laboratory findings of patients with non-infectious uveitis and controls.

	BD	Sarcoidosis	VKH	Control
Number	19	20	15	16
Sex (male/female)	9/10	6/14	8/7	8/8
Age (year) ± SD (range)	42.7 ± 16.8 (15–71)	69.0 ± 11.3 (45–86)	46.4 ± 15.5 (12–71)	53.8 ± 18.5 (27–83)
WBC (/μL)	6307.5	6204.2	6508.3	-
CRP (mg/dL)	0.11	0.09	0.08	-
CH50 (U/mL)	62.1	63.8	59.1	-
sIL-2R (U/mL)	410.3	810.2	-	-
ACE (IU/L)	12.1	18.4	9.2	-
HLA-B51	7 (36.8%)	-	-	-
HLA-A26	8 (42.1%)	-	-	-

BD: Behҫet’s disease, VKH:Vogt-Koyanagi-Harada disease, SD: standard deviation, WBC: white blood cell, CRP: C-reactive protein, CH50: 50% hemolytic unit of complement, sIL-2R: soluble IL-2 receptor, ACE: angiotensin-converting enzyme, HLA: Human Leukocyte Antigen.

**Table 2 jcm-09-03955-t002:** Statistics of MLR model.

Metabolite	Parameters	95% CI		*p*-Value	Odds Ratio	95% CI	
(Sarcoidosis + BD + VKH) from C							
(Intercept)	0.614	−0.617	1.40	0.123			
Adenosine	−39.6	−78.1	−1.06	0.0440	6.44 × 10^−18^	1.20 × 10^−34^	0.346
Nicotinamide	3.41	0.469	6.35	0.0230	30.2	1.60	1.20 × 10^2^
Sarcoidosis from (BD + VKH)							
(Intercept)	0.916	−2.21	4.04	0.566			
Thiamine	16.2	6.3	26.2	1.40 × 10^−3^	1.14 × 10^7^	5.46 × 10^2^	2.37 × 10^11^
Asparagine	−0.0812	−0.157	−5.06 × 10^−3^	0.0366	0.922	0.854	0.995
BD from (Sarcoidosis + VKH)							
(Intercept)	−0.810	−2.25	0.633	0.271			
Thiamine	−14.6	−25.3	−3.91	7.40 × 10^−3^	4.51 × 10^−7^	1.02 × 10^−11^	0.0200
Uridine	0.801	0.215	1.39	7.40 × 10^−3^	2.23	1.24	4.00
VKH from (BD + Sarcoidosis)							
(Intercept)	1.67	−0.541	3.88	0.139			
Citrulline	−0.0947	−0.176	−0.0134	0.0244	0.91	0.839	0.987

MLR: Multiple logistic regression, C: Control, BD: Behҫet’s disease, VKH: Vogt-Koyanagi-Harada disease, CI: confidence interval.

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
