# Peer review of "Serum Metabolomic Profiling of Patients with Non-Infectious Uveitis"

_jcm, 2020, doi:10.3390/jcm9123955_

Round 1

Reviewer 1 Report

The authors present an elegant paper, describing the use of metabolomics analysis to discriminate between 3 different subsets of non-infectious uveitis.

  • In Fig 1: Can the authors clarify, more details about the control group. The authors state that “16 patients with no history of autoimmune disease or malignant tumor were treated as controls”. Did these samples come from patients with other know diseases? Or were they truly healthy controls?
  • Additionally, can the authors confirm, that the serum sample analyzed was taken at a time point when the patient was active.
  • Although patients with systemic corticosteroids were excluded, were any of the patients on any local treatments which might affect the metabolomic profile? E.g. AC cells, Vitreous Haze, etc.
  • Although there is some clinical characteristics, are there any uveitis specific clinical characteristics, which corelate with the metabolomic findings?
  • The authors beautifully show differences between, the three grips of patients. However, fewer differences were noted in the control group vs all the three patient groups combined. Can the authors talk about this discrepancy in the discussion?
  • The three subtypes of uveitis analyzed are distinct etiologies. However, there are still a substantial number of patients who are classified as idiopathic uveitis. Will such an analysis, be able to shed more light on this group?
  • Maybe adding PCA plots, might help better visual the differences between the groups.

Author Response

The authors present an elegant paper, describing the use of metabolomics analysis to discriminate between 3 different subsets of non-infectious uveitis.

Response

We thank the reviewer for the thoughtful and encouraging comments.

In Fig 1: Can the authors clarify, more details about the control group. The authors state that “16 patients with no history of autoimmune disease or malignant tumor were treated as controls”. Did these samples come from patients with other know diseases? Or were they truly healthy controls?

Response

We thank the reviewer for the comment. The 16 control subjects comprised 7 cataract patients with no history of autoimmune disease or malignant tumor and 9 healthy subjects with no history of autoimmune disease or malignant tumor. We have added these details in 2.1 Patients and diagnosis as follows:

“Sixteen subjects (7 cataract patients and 9 healthy subjects) with no history of autoimmune disease or malignant tumor were included as controls.” (page 2 line 91)

Additionally, can the authors confirm, that the serum sample analyzed was taken at a time point when the patient was active.

Response

We thank the reviewer for the comment. Yes, we confirmed that the patient had active disease at the time when serum samples were collected for analysis. We have added this information in 2.3 Serum sample collection as follows:

“The patient were confirmed to have active disease at the time when serum samples were collected for analysis.” (page 3 lines 103-104)

Although patients with systemic corticosteroids were excluded, were any of the patients on any local treatments which might affect the metabolomic profile? E.g. AC cells, Vitreous Haze, etc.

Response

Since our hospital is a referral center, some patients at the time of referral were receiving topical steroids or mydriatics. These patients were included for analysis, since it is impossible to select totally treatment naïve patients. To address the comments of two reviewers, we have added this information in 2.1 Patients and diagnosis as follows:

“Patients who had been treated with topical steroids and mydriatics were included for analysis.” (page 2 lines 89-90)

We have stated this as a limitation in 4. Discussion as follows:

“Although patients who had been treated with systemic corticosteroids within the past 6 months were excluded, some patients included in the study had been treated with local therapies such as topical steroids and mydriatics, which could affect the metabolomic profile.” (page 8 lines 287-290)

Although there is some clinical characteristics, are there any uveitis specific clinical characteristics, which corelate with the metabolomic findings?

Response

We thank the reviewer for the comment. We have examined the correlation between clinical characteristics (including WBC, sIL-2R, and ACE) and metabolites (adenosine, nicotinamide, thiamine, asparagine, uridine and citrulline) shown to be significantly different among diseases shown in Table 2, as well as some clinical characteristics (including WBC, sIL-2R, and ACE) shown in Table 1. However, we did not find any significant correlation.

The authors beautifully show differences between, the three grips of patients. However, fewer differences were noted in the control group vs all the three patient groups combined. Can the authors talk about this discrepancy in the discussion?

Response

We thank the reviewer for the comment. Indeed, if we look at Fig. 3E, there are little differences when comparing the group combining all the three diseases vs the control group. Although BD, sarcoidosis and VKH are all uveitis, they are completely different diseases. When we combine the three together, this highly heterogenous group likely exhibits a highly diverse metabolic profile, and when compared with the control group results in little differences between the two groups. We have discussed this point in the revised manuscript as follows:

“Although BD, sarcoidosis and VKH are all uveitis, they are completely different diseases. When we combine the three together in one group, this highly heterogeneous group likely exhibits a diverse metabolomic profile, resulting in low discrimination when compared with the control group. When the three uveitis diseases were compared with each other, their distinct metabolomic profiles probably increase the discrimination sensitivity among the three diseases (Figure 3).” (page 7 lines 238-242)

The three subtypes of uveitis analyzed are distinct etiologies. However, there are still a substantial number of patients who are classified as idiopathic uveitis. Will such an analysis, be able to shed more light on this group?

Response

We thank the reviewer for the important comment. We believe that metabolome analysis of idiopathic uveitis is very important, and will shed light on this this group of uveitis. Furthermore, metabolome analysis may help elucidate novel pathogenesis of this disease. We have commented on this point in Discussion as follows:

“Especially, metabolomic profiling of idiopathic uveitis would be important for elucidation of novel pathogenesis of this entity and for discrimination from the three major uveitis.” (page 8 lines 297-298)

Maybe adding PCA plots, might help better visual the differences between the groups.

Response

We attempted to construct PCA plots (Please find the Word file), but they do not improve visual differences between the groups. This may be due to the difference in analytical methods. Since we found little differences in PCA, we conducted multiple logistic regression analysis using a combination of only a few factors that showed significant differences. 

Reviewer 2 Report

This is an interesting article presenting a promising technique that might be useful in uveitis evaluation, improving our understanding of relevant pathophysiological mechanisms of disease, and potentially leading to improved treatment; efforts on such an important topic should be lauded.

However, a very important aspect should be clarified: in line 84 it is said that patients had “active uveitis” and that “Patients who had been treated with systemic corticosteroids within the past 6 months leading up to the acquisition of the serum sample were excluded”; it must be clearly stated which systemical or topical treatments were being used; in fact, this is a fundamental aspect in this paper, as different treatments may have altered the Authors' results, affecting metabolite concentrations in the patients’ sera; moreover this should be added as a major limitation of this work in case different treatments were being used.

Reference 21 should be formatted according to the Journal guidelines.

Author Response

Reviewer#2

This is an interesting article presenting a promising technique that might be useful in uveitis evaluation, improving our understanding of relevant pathophysiological mechanisms of disease, and potentially leading to improved treatment; efforts on such an important topic should be lauded.

Response

We thank the reviewer for the thoughtful and encouraging comments.

However, a very important aspect should be clarified: in line 84 it is said that patients had “active uveitis” and that “Patients who had been treated with systemic corticosteroids within the past 6 months leading up to the acquisition of the serum sample were excluded”; it must be clearly stated which systemical or topical treatments were being used; in fact, this is a fundamental aspect in this paper, as different treatments may have altered the Authors' results, affecting metabolite concentrations in the patients’ sera; moreover this should be added as a major limitation of this work in case different treatments were being used.

Response

We thank the reviewer for the pertinent comment. The treatments that patients were receiving are indeed important as they may affect the metabolomic profiles. As our hospital is a referral center, some patients at the time of referral were receiving topical steroids or mydriatics. These patients were included for analysis, since it is impossible to select totally treatment naïve patients. To address the same comments from both reviewers, we have added the information of treatment in 2.1 Patients and diagnosis as follows:

“Patients who had been treated with systemic corticosteroids within the past 6 months leading up to the acquisition of the serum sample were excluded. Patients who had been treated with topical steroids and mydriatics were included for analysis.” (page 2 lines 89-90)

We have stated this as a limitation in 4. Discussion as follows:

“Although patients who had been treated with systemic corticosteroids within the past 6 months were excluded, some patients included in the study had been treated with local therapies such as topical steroids and mydriatics, which could affect the metabolomic profile.” (page 8 lines 287-290)

Reference 21 should be formatted according to the Journal guidelines.

Response

We apologize for the formatting error. We have changed the format according to Journal guidelines.

Round 2

Reviewer 2 Report

The Authors have modified their paper, but it must be clearly stated which systemical treatments were being used in these patients with active uveitis (antimetabolites ? T-cell inhibitors like cyclosporine ? ...); in fact, this is a fundamental aspect in this paper, as different treatments may have altered the Authors' results, affecting metabolite concentrations in the patients’ sera; this should be added as a major limitation of this work in case different treatments were being used.

Author Response

The Authors have modified their paper, but it must be clearly stated which systemical treatments were being used in these patients with active uveitis (antimetabolites ? T-cell inhibitors like cyclosporine ? ...); in fact, this is a fundamental aspect in this paper, as different treatments may have altered the Authors' results, affecting metabolite concentrations in the patients’ sera; this should be added as a major limitation of this work in case different treatments were being used.

Response: We thank the review for appraising our revision and for the comment. We have already stated in our revised manuscript that patients who had been treated with systemic corticosteroids within the past 6 months leading up to the acquisition of the serum sample were excluded (P2L87-89).

To avoid misunderstanding, we have added the following:

“Patients who had been treated with systemic corticosteroids, immunosuppressive drugs, and antimetabolites within the past 6 months leading up to the acquisition of the serum sample were excluded. (P2 L87-89).